# Temporal and Wash-Out Studies Identify Medicines for Malaria Venture Pathogen Box Compounds with Fast-Acting Activity against Both *Trypanosoma cruzi* and *Trypanosoma brucei*

**DOI:** 10.3390/microorganisms10071287

**Published:** 2022-06-25

**Authors:** Melissa L. Sykes, Emily K. Kennedy, Kevin D. Read, Marcel Kaiser, Vicky M. Avery

**Affiliations:** 1Discovery Biology, Griffith University, Nathan, QLD 4111, Australia; emily-kate.91@hotmail.com (E.K.K.); v.avery@griffith.edu.au (V.M.A.); 2Wellcome Centre for Anti-Infective Research, Drug Discovery Unit, Division of Biological Chemistry and Drug Discovery, School of Life Sciences, University of Dundee, Dundee DD1 5EH, UK; k.read@dundee.ac.uk; 3Swiss Tropical and Public Health Institute (Swiss TPH), 4123 Allschwil, Switzerland; marcel.kaiser@swisstph.ch; 4University of Basel, 4002 Basel, Switzerland; 5School of Environment and Science, Griffith University, Nathan, QLD 4111, Australia

**Keywords:** MMV Pathogen Box, Human African Trypanosomiasis, Chagas disease, *Trypanosoma cruzi*, *Trypanosoma brucei*, image-based assays, high-content imaging, speed of action, time-kill assay, cidal activity, *T. cruzi* cytochrome P450, antitrypanosomal

## Abstract

Chagas disease caused by the protozoan *Trypanosoma cruzi* is endemic to 21 countries in the Americas, effects approximately 6 million people and on average results in 12,000 deaths annually. Human African Trypanosomiasis (HAT) is caused by the *Trypanosoma brucei* sub-species, endemic to 36 countries within sub-Saharan Africa. Treatment regimens for these parasitic diseases are complicated and not effective against all disease stages; thus, there is a need to find improved treatments. To identify new molecules for the drug discovery pipelines for these diseases, we have utilised in vitro assays to identify compounds with selective activity against both *T. cruzi* and *T.b. brucei* from the Medicines for Malaria Venture (MMV) Pathogen Box compound collection. To prioritise these molecules for further investigation, temporal and wash off assays were utilised to identify the speed of action and cidality of compounds. For translational relevance, compounds were tested against clinically relevant *T.b. brucei* subspecies. Compounds with activity against *T. cruzi* cytochrome P450 (TcCYP51) have not previously been successful in clinical trials for chronic Chagas disease; thus, to deprioritise compounds with this activity, they were tested against recombinant TcCYP51. Compounds with biological profiles warranting progression offer important tools for drug and target development against kinetoplastids.

## 1. Introduction

The kinetoplastid borne diseases, Human African Trypanosomiasis (HAT) and Chagas disease, threaten the lives of over 90 million people in 56 countries, spread over 3 continents. Issues associated with treatment regimens, toxicity and efficacy of drugs for the treatment of these diseases are considerable. Chagas disease is categorised by the World Health Organisation (WHO) as one of the world’s 20 most neglected tropical diseases (NTDs) [1]. The disease is endemic to 21 countries within the Americas, and causes more than 12,000 deaths per year, with a further 75 million people at risk of becoming infected with *T. cruzi* [2]. Current drugs used to treat Chagas disease, benznidazole and nifurtimox, lack efficacy in the chronic stage of disease and have associated side effects, resulting in reduced compliance [3]. In addition, treatment times for both drugs are lengthy, requiring 60–90 days [4]. With few compounds currently in clinical trials for Chagas disease, seeding the drug discovery pipeline with new and efficacious compounds is necessary in the search for new therapies with improved safety profiles and reduced treatment schedules.

HAT is caused by two subspecies, *Trypanosoma brucei gambiense* and *Trypanosoma brucei rhodesiense*, and presents in two stages, whereby parasites cross the blood–brain barrier in the second stage. None of the drugs used to treat HAT are effective against all disease stages and parasite species. However, the first oral drug, fexinidazole, was approved for treatment of both stages of HAT caused by *T.b. gambiense* in 2019 [5]. The Drugs for Neglected Diseases initiative (DND*i*) is partnering with an African-based consortium (HAT-r-ACC) to undertake a 5-year study to assess the efficacy of fexinidazole against *T.b. rhodesiense* [6] and clinical trials are underway [7]. Currently, the only drug approved for stage-two HAT caused by *T.b. rhodesiense* is melarsoprol. Melarsoprol is very effective against HAT; however, post-treatment reactive encephalopathy is reported to occur in 5–10% of all treated cases and is fatal in 50% of those patients in which encephalopathy develops [8]. Whilst there is one other compound in the pre-clinical pipeline, acoziborole, it is again for the treatment of *T.b. gambiense.* Given that high attrition rates in drug discovery are well-documented, with lack of efficacy and safety issues as the main causes [9], new orally available compounds for the safe treatment of HAT are needed. 

Medicines for Malaria venture (MMV) collated the Pathogen Box, containing 400 diverse, drug-like molecules active against the causative agents of various neglected tropical diseases (NTDs). This includes, but is not limited to, mycobacterial species causing tuberculosis and parasites causing schistosomiasis, HAT, Chagas disease and leishmaniasis [10]. This open-source collection was established to accelerate NTD drug discovery, especially those affecting low socioeconomic groups which disproportionally affect developing countries [10,11]. Open-source drug discovery using the Pathogen Box has successfully identified lead compounds for pathogens such as *Shistosoma mansoni* (schistosomiasis); *Candida albicans* (candidiasis), piroplasm parasites (including *Babesia* sp) and *Fasciola hepatica* (fascioliasis) [12,13,14,15]. In our previous in vitro study, the Pathogen Box was profiled against *T. cruzi* and *T.b. brucei*, in addition to *Plasmodium falciparum* and *Leishmania donovani* [16]. Forty-three compounds selectively active against either *T. cruzi* or *T.b. brucei* were identified that previously were not shown to be active against theses parasites; or if reported as active, have an undefined target [16]. From the 43 compounds, 93% had an undefined target and 65% had not been reported with activity against either one or both parasites. Sixty-eight percent of these have since been reported as active against *T.b. brucei* [17]; however, the target and dual activity against both *T.b. brucei* and *T. cruzi* was not reported. To confirm activity against *T. cruzi* and *T.b. brucei*, and prioritise compounds for drug discovery efforts, in vitro profiling to classify rapidly acting compounds with irreversible activity against *T. cruzi* and activity against the infective forms of HAT was undertaken with solid compound stocks. The speed of action and efficacy against both parasites were identified.

The speed of compound action is an important consideration in drug discovery, as this often reflects potential treatment duration, important from the perspective that elongated treatment regimens are an issue for both Chagas disease and HAT [2,18]. For Chagas disease, slow growing/persistent parasites are considered of particular clinical relevance [19,20]. A four-fold difference has been found in *T. cruzi* load in acute versus chronic infections in mice [21], suggesting that replication of the parasite may be reduced, limited, or prevented in vivo. In vivo analysis of mouse models of infection has recently shown *T. cruzi* parasites to persist in low numbers, in gut tissue [22]. Slow growing parasites may be resistant to treatment by compounds that do not rapidly kill; thus, we utilised an image-based rate of kill assay to assess the speed of action of compounds against *T. cruzi* [23]. Time-kill assays, that we have previously established for *T.b. brucei* [24], were utilised to assess the speed of action of compounds against *T.b. brucei* in this study. 

A cidal mode of action (MOA) indicates the compound effect on the target cell is not reversible, or there is a commitment to apoptosis [25]. Current wash-out assays developed to identify the cidal nature of compounds against *T. cruzi* are lengthy and, considering the long incubation periods up to 60 days [26,27], the health of host cells may be compromised. We have developed a short-term wash off assay to identify compounds with an irreversible effect on *T. cruzi* parasites, by incubating in the presence of compound for 48 h [28], followed by incubation in the absence (wash off) of compound for 72 h [23]. This method enabled the rapid assessment of the cidality of compounds and thus the prioritisation of active compounds from the Pathogen Box, with a potentially irreversible MOA against *T. cruzi*. 

*T. cruzi* cytochrome P450 (TcCYP51) inhibitors, such as posaconazole, have not been successful in clinical trials for the treatment of chronic Chagas disease [29,30]. Whilst the lack of efficacy of posaconazole may have been influenced by the concentration and duration of exposure in these clinical trials, currently posaconazole cannot be considered as a monotherapy [31]. As a result, many programs in *T. cruzi* drug discovery deprioritise CYP51 inhibitors [25,27]. Following 48 h exposure of *T. cruzi* infected cells to posaconazole there is incomplete clearance of *T. cruzi* parasites [23,32]. We have utilised temporal image-based assays to assess active compounds from the Pathogen Box collection against *T. cruzi*, to deprioritise a CYP51-like phenotype, which typically shows low efficacy following 48 h incubation and a slow-acting MOA [23]. In this current work, we have confirmed the lack of activity of prioritised compounds against *T. cruzi* CYP51, utilising the *T. cruzi* recombinant enzyme [33].

Compounds with selective activity against both parasites, which do not target TcCYP51 with an irreversible MOA, and exhibit activity against the human infective forms of *T. brucei* serve as promising hits for Chagas disease and HAT drug discovery. These compounds provide tools to probe parasite biology via target identification. Recently, there was hope with proteasome inhibitors for developing treatments against multiple kinetoplastids [34], as there are similarities between DNA repair pathways for these parasites [35]. Fexinidazole, used to treat *T.b. gambiense* infection [36], has also recently been in clinical trials for Chagas disease [37], indicating the potential that compounds of the same compound class can be pursued as new chemical starting points for both parasites. 

## 2. Methods

### 2.1. Growth and Maintenance of Parasites and Host Mammalian Cell Lines

*T. cruzi* Tulahuen strain parasites and 3T3 host cells (ATCC) were maintained as outlined previously [28]. Briefly, host 3T3 mouse fibroblasts (ATCC CCL-92) were grown in RPMI medium (ThermoFisher, Waltham, MA, USA) with no phenol red, supplemented with 10% FBS (ThermoFisher, Waltham, MA, USA) and incubated at 37 °C in 5% CO_2_. Host cells were harvested at 70% confluency and split every 2–3 days in 175 cm^2^ flasks. 3T3 cells are contact inhibited and thus were cultured up to passage 7 to minimise overgrowth of culture. Prior to infection with *T. cruzi* trypomastigotes, host cells were seeded at 1.2 × 10^6^ cells in 75 cm^2^ or 4 × 10^5^ cells in 25 cm^2^ flasks for 24 h before parasite addition. Parasites were added at a 10:1 multiplicity of infection (MOI), incubated for 24 h and extracellular trypomastigotes washed off the host cell bed with PBS supplemented with Ca^2+^ and Mg^2+^ (ThermoFisher, Waltham, MA, USA). Incubation was continued for a further 72 h, until tissue culture trypomastigotes initiated egress from host cells. Parasites were harvested from the flask supernatant and used for assays, or to continue the infective life cycle in vitro. 

*T.b. brucei* bloodstream 427 strain parasites were sub-cultured as previously reported in HMI-9 medium [24], supplemented with 10% FBS. Parasites were maintained in the log phase of growth during sub-culturing for 48 or 72 h intervals, at 37 °C and 5% CO_2_ in 25 cm^2^ filtered tissue culture flasks (Corning, New York, NY, USA), under humidified conditions.

Human embryonic kidney cells (HEK293; ATCC) were grown in high glucose DMEM supplemented with 10% FBS at 37 °C and 5% CO_2_. Cells were sub-cultured every 3 or 4 days by seeding 1.5 × 10^6^ or 1 × 10^6^ cells, respectively, into 175 cm^2^ flasks. Human hepatocellular carcinoma cells (HEPG2; ATCC) were grown in high glucose DMEM supplemented with 10% FBS at 37 °C and 5% CO_2_. Cells were sub-cultured every 3 or 4 days by seeding 1.5 × 10^6^ or 1 × 10^6^ cells, respectively, into 175 cm^2^ flasks. 

### 2.2. Compounds

All compounds were prepared in 100% DMSO and diluted in 14 log dilutions to determine IC_50_ values. Reference compounds for the inhibition of *T.b. brucei* growth following 24, 48 and 72 h were diminazene aceturate, with final assay concentrations ranging from 38 µM to 1.9 × 10^−3^ µM and puromycin, an inhibitor of eukaryotic protein synthesis [38] as a control for general cytotoxicity with final assay concentrations ranging from 36 µM to 1.8 × 10^−3^ µM (Sigma Aldrich, Saint Louis, MO, USA). The positive control, diminazene, was at a final assay concentration of 19 µM (EC_100_) and the negative control was 0.4% DMSO (vehicle only). Reference compounds for the inhibition of *T. cruzi* growth were benznidazole ranging from 127 µM to 6.3 × 10^−3^ µM (supplied by EpiChem Pty Ltd., Bentley, WA, Australia); nifurtimox ranging from 127 µM to 6.3 × 10^−3^ µM (isolated from Lampit tablets by Dr. Agatha Garavalis), posaconazole ranging from 1 µM to 5 × 10^−5^ µM (Sigma Aldrich, Saint Louis, MO, USA) and puromycin from 200 µM to 1 × 10^−2^ µM (Sigma Aldrich, Saint Louis, MO, USA) as a control of general cytotoxicity. The positive control, nifurtimox, was at a final assay concentration of 12 µM (EC_100_) and the negative control was 0.4% DMSO (vehicle only).

Forty-three compounds from the MMV Pathogen Box library were obtained as 1 mg solid stocks from MMV. These compounds previously identified with activity against *T.b. brucei* and *T. cruzi* in our previous publication [16]. Compounds were either potential hit molecules or probes for identification of new targets, for *T. cruzi* and *T.b. brucei* parasites. Prior to use, compounds were prepared in 100% DMSO at a stock concentration of 20 mM.

### 2.3. T.b. brucei 72 h REDOX-Based Viability Assay

The *T.b. brucei* 72 h in vitro viability assay was undertaken as previously described [16]. Briefly, parasites were inoculated into sterile polystyrene black/clear bottomed 384-well plates (Greiner, Monroe, NC, USA) at 1.2 × 10^3^ cells/ mL in 55 µL of HMI-9, supplemented with 10% FBS. Parasites were incubated for 24 h prior to the addition of 5 µL of pre-diluted control or test compounds. Plates were incubated for 48 h before addition of 10 µL of 490 µM resazurin (Cayman Chemicals, Ann Arbor, MI, USA), to give a final concentration of 70 µM. Plates were incubated for 22 h at 37 °C, 5% CO_2_ followed by incubation at room temperature for 2 h. Total well fluorescence was determined using an EnSight plate reader (PerkinElmer, Waltham, MA, USA) at an excitation of 535 nm and emission of 590 nm. Compound additions to plates were performed with a MiniTrak liquid dispenser (PerkinElmer, Waltham, MA, USA). Compounds in 100% DMSO were pre-diluted in high glucose DMEM (ThermoFisher, Waltham, MA, USA) at a ratio of 1:21, before addition to assay plates. For dose response of control compounds, concentrations ranged from 38 to 1.9 × 10^−3^ µM for diminazene and 36 to 1.8 × 10^−3^ µM for puromycin. The positive control, diminazene, was used at a final assay concentration of 19 µM (EC_100_) and the negative control was 0.4% DMSO (vehicle only). A plate containing 14 doses of control compounds as described was included in every assay, with two biological replicates for each compound concentration series. In addition, negative (0.4% DMSO) and positive (puromycin and diminazene) in plate-controls (EC_100_) were included for each assay plate containing test compounds, with one column for each control. The Z′-factor was calculated from negative and positive controls to evaluate the reproducibility of the assay, with values > 0.5 indicating a reproducible assay [39]. 

### 2.4. T. cruzi 48 h Image-Based Assay 

The *T. cruzi* 48 h image-based assay was performed as previously described [28]. Briefly, 3T3 mouse fibroblast cells were added at 1 × 10^3^ cells/well in 50 μL of medium into 384-well Collagen I coated plates (CellCarrier, PerkinElmer, Waltham, MA, USA) using a Multidrop (ThermoFisher Scientific, Waltham, MA, USA). Plates were incubated for 24 h at 37 °C and 5% CO_2_ before addition of 10 μL of *T. cruzi* trypomastigotes at a MOI of 5:1 parasite–host cells. Plates were incubated for 24 h and then wells were washed three times with PBS containing magnesium and calcium, once with additions using a Multidrop (ThermoFisher Scientific, Waltham, MA, USA) and twice using a Bravo liquid handler (Agilent Technologies, Santa Clara, CA, USA), with 50 μL of media replaced after the third wash. The initial wash was carried out by discarding the supernatant into a reservoir containing bleach, under sterile conditions. Five microlitres of prediluted compounds were added to plates with a MiniTrak compound handing device (PerkinElmer, Waltham, MA, USA) and incubated for 48 h before fixing cells with paraformaldehyde and staining with fluorescent nuclear and cytoplasmic markers. Plates were imaged on an Opera image-based system (PerkinElmer, Waltham, MA, USA). 

### 2.5. Pathogen Box IC_50_ Values and Parasite Selectivity 

The 43 compounds from the Pathogen Box were tested over 14 log doses ranging from final assay concentrations of 79 µM to 4.0 × 10^−3^ µM against *T.b. brucei* and HEK 293 cells, and from 73 µM to 3.7 × 10^−3^ µM for *T. cruzi* and 3T3 host cells. Confirmed selective compounds to parasites in relation to HEK293 cells for either parasite progressed to temporal and wash-out assays and activity was determined against HEPG2 cells. Compounds were tested in two independent experiments (*n* = 2). Compounds with fast-acting activity against both parasites were prioritised for testing against *T.b. brucei* subspecies and TcCYP51.

### 2.6. T.b. brucei Temporal Viability Assays

Time-kill assays was undertaken as previously described, using the REDOX reagent PrestoBlue^®^ [24]. To investigate the influence of 24 h incubation with compounds, plates were incubated for 15 h before the addition of 10 µL of 6× PrestoBlue^®^ (ThermoFisher, Waltham, MA, USA), diluted in HMI-9 supplemented with 10% FBS, to give a final concentration of 1×. Plates were incubated for 9 h, giving a total incubation time of 24 h after addition of compounds. Control compounds were added at the same concentrations at outlined for the resazurin-based assays.

To investigate the effect of 48 h exposure to compounds on parasite viability, *T.b. brucei* were incubated with compound for 45 h followed by addition of Presto Blue (final 1× concentration) and further incubation for 3 h at 37 °C and 5% CO_2_. After either 24 h or 48 h incubation with compound and addition of Presto Blue, plates were read on an EnSight plate reader (PerkinElmer, Waltham, MA, USA), at Ex/Em 535/590 nm.

### 2.7. T. cruzi Temporal Image-Based Assays

Assays to assess the activity of compounds following 72 h exposure [23,28] and 24 h exposure [23] to *T. cruzi* infected host cells were undertaken as previously described. Briefly, the method for the 48 h assay [28] was modified for either 24 or 72 h. In the 72 h assay, the number of wash cycles was increased to 5 instead of 2 during PBS wash steps to ensure removal of parasites that had egressed from host cells. 

### 2.8. T.b. brucei Subspecies Assays 

The assays to determine the sensitivity of *T. brucei* human infective subspecies to prioritised Pathogen Box compounds were undertaken at the Swiss Tropical and Public Health Institute (Swiss TPH, Allschwil, Switzerland). The *T.b. rhodesiense* STIB900 and *T.b. gambiense* STIB930 assays were performed as previously described, with compound exposure for 72 h [24,40]. 

### 2.9. HEK293 Cytotoxicity Assay 

The HEK293 cell cytotoxicity assay was carried out as described previously [24], modified slightly with the use of resazurin. Briefly, HEK293 cells (ATCC) were added to sterile polystyrene black/clear bottomed 384-well plates (Greiner, USA) at 4 × 10^3^ cells per well, in 55 µL of high glucose DMEM supplemented with 10% FBS. Following 24 h incubation at 37 °C in 5% CO_2_, under humidified growth conditions, 5 µL of pre-diluted compound was added with a MiniTrak (PerkinElmer, Waltham, MA, USA). Final assay concentrations of Pathogen Box compounds ranged from 79.4 µM to 4.0 × 10^−3^ µM. Plates were incubated for 68 h, then 10 µL of 490 µM of resazurin was added to give a final concentration of 70 µM. Plates were incubated for a further 4 h under normal growth conditions giving a total compound incubation of 72 h. Fluorescence (Ex/Em 535/590 nm) was then determined on an EnSight plate reader (PerkinElmer, Waltham, MA, USA). Positive and negative controls used were as the same as those in the *T.b. brucei* assay; 4 µM puromycin was a positive control and 0.4% DMSO served as a negative control. 

### 2.10. HEPG2 Cytotoxicity Assay

The assay was undertaken as described for the HEK293 assay, with a modification of the cell number used to 500 cells/well. Five microlitres of compounds pre-diluted 1:21 in DMEM were added to plates to give final assay concentrations between 79.4 µM to 4.0 × 10^−3^ µM. Plates were incubated for 68 h, before the addition of 10 µL of 490 µM of resazurin to give a final concentration of 70 µM. Plates were incubated for a further 4 h prior to being read on an EnSight plate reader (PerkinElmer, Waltham, MA, USA) at an excitation of 535 nm and emission of 590 nm. The concentration of the control compounds utilised for this assay were the same as those prepared for the *T.b. brucei* 72 h assay.

### 2.11. Compound Activity against T. cruzi Recombinant CYP51 Enzyme

The inhibitory properties of compounds against *T. cruzi* recombinant CYP51 (TcCYP51) was determined using an in vitro assay as previously described [33,41]. Compounds which inhibited *T. cruzi* in an image-based assay were tested against TcCYP51 in log concentrations (100, 33, 10, 3.3, 1.0, 0.33, 0.1 and 0 µM), with one replicate per dose, as previously described [41]. The control compound posaconazole was incubated at the same concentrations as the test compounds, over two replicates. 

## 3. Results

### 3.1. Confirming Activity of Pathogen Box Solid Compound Stocks against T.b. brucei and T. cruzi

New solid stocks of the 43 compounds from the Pathogen Box were tested to determine activity against both parasites, with the selectivity of active compounds determined for HEK293 and HEPG2 cells (for those compounds with selectivity against HEK293), in addition to 3T3 host cells in the *T. cruzi* assay. Active compounds were defined as those with an IC_50_ value of <10 µM [16]. Of the 43 compounds tested, 19 were originally found selectively active against *T.b. brucei*; 14 compounds active against both *T.b. brucei* and *T. cruzi* and 10 active only against *T. cruzi* [16]. From the compounds active against *T.b. brucei* (either selectively, or active against both *T.b. brucei* and *T. cruzi*), 27 reconfirmed activities against *T.b. brucei* (82% reconfirmation: 27 from 33 compounds). From compounds that were active against *T.b. brucei* only from the original study [16], 8 compounds were found to also be active against *T. cruzi* upon testing the solid stocks. Table 1 shows that 22 compounds from the 43 compounds profiled demonstrated activity against both *T.b. brucei* and *T. cruzi*. Of these, MMV022029, MMV022478, MMV688514 and MMV21013 were not selective against either parasite in relation to HEK293 cells (SI < 10) and displayed activity against 3T3 cells (Table 1), and thus were deprioritised. MMV006901 did not display selectivity to *T.b. brucei* and showed variable activity against 3T3 cells. This compound was subsequently tested against HEPG2 and did not show selectivity for either parasite; therefore, this was deprioritised. MMV687706 had activity above the cut off of 10 µM against *T. cruzi* (11 µM) and was not selective to *T.b. brucei* (SI = 7.4) and thus was not progressed. MMV028694 demonstrated poor selectivity to *T. cruzi* in relation to HEK293 cells (SI = 1); however, it was selective against *T.b. brucei*, and *T. cruzi* to 3T3 cells, and thus was progressed to further in vitro profiling.

MMV028694 and MMV688283 were moderately activity against *T. cruzi* (9.4 ± 1.8 µM and 6.6 ± 1.2 µM, respectively, Table 1) and this activity may not have been identified with the lower concentration of 20 µM used in our previous testing [16], whilst MMV099637 and MMV688776 gave IC_50_ values of 2.7 µM and 3.2 µM, respectively (Table 1), despite not being identified as active against *T. cruzi* in our previous study using the liquid stocks provided. From the original Pathogen Box testing, 13 compounds were selectively active against *T. cruzi* and *T.b. brucei* parasites, with 13/14 confirming activity (93%) against *T. cruzi*. The compound that did not confirm activity was MMV687273 (SQ-109, Tuberculosis set), which was less active than the previous test, displaying 16 µM (results not shown) versus 5.4 µM in the previous studies [16].

*T.b. brucei* and HEK293 assays gave mean Z′ values of 0.87 and 0.84, respectively, and *T. cruzi* and 3T3 assays mean Z′-values of 0.56 and 0.75, respectively, indicating reproducible assays. 

### 3.2. Time-Kill: Compounds Active against Both T. cruzi and T.b. brucei

Compounds selectively active against both parasites were tested for temporal activity against *T. cruzi* and *T.b. brucei*, and regrowth following the wash-out of compounds in the *T. cruzi* assay (Table 2). Compounds were categorised as fast-acting if they had an IC_50_ value of <10 µM following 24 h incubation, with a maximum inhibition on the plateau of activity (E_max_) of ≥80%. The E_max_ was determined as previously described [23]. Compounds were categorised cidal if the E_max_ was >90% following compound wash off, and there was no decrease in the E_max_ between 48 h incubation (<10% difference in the % E_max_) and following wash off of compound (48 h incubation, followed by wash off of compound, then a further 72 h incubation). Compounds were defined as efficacious if they cleared >90% *T. cruzi* parasites following 48 h exposure, following our previous criteria for definition of sub-efficacious compounds against *T. cruzi*. This profile may indicate TcCYP51 inhibition [23].

### 3.3. Fast-Acting Compound Activity against T.b. rhodesiense and T.b. gambiense

Compounds that were fast-acting and efficacious against both *T. cruzi* and *T.b. brucei* were tested against the human infective forms of *T. brucei.* The IC_50_ values of these compounds against human infective *T. brucei* spp. are shown in Table 3.

### 3.4. Physicochemical Properties of Fast-Acting Compounds

The physicochemical properties of the fast-acting compounds identified with activity against *T. cruzi* and *T.b. brucei* are shown in Table 4. The molecular weight and LogP (XlogP) of test compounds were sourced from PubChem (https://pubchem.ncbi.nlm.nih.gov/, accessed on 27 May 2022) and control compounds from DrugBank (https://www.drugbank.com/, accessed on 27 May 2022). All compounds have a LogP value of <5 and a molecular weight of <500 and thus are predicted to have favourable absorption and permeation in vivo, according to Lipinski [43].

### 3.5. TcCYP51 Biochemical Activity

Slow- and fast-acting compounds active against both *T. cruzi* and *T.b. brucei* were tested for their ability to inhibit TcCYP51 enzymatic activity (Figure 1). Of the fast-acting compounds, if there was more than one compound with activity against *T. cruzi*, one representative compound from each chemical class was selected. MMV652003 was not tested as benzoxaboroles have known activity against the parasite. Slow-acting compounds were also tested for activity against TcCYP51. The criteria defining a compound with TcCYP51 activity required that the IC_50_ value was within a log difference to posaconazole, as previously reported [33]. None of the compounds tested showed inhibition of TcCYP51, based on these criteria.

### 3.6. Compounds Recommended for Further Investigation

Prioritised compounds recommended for further investigation based on in vitro profiling and literature searches are shown in Table 5. Increased SAR exploration is required for the remainder of the compounds to identify compounds with improved selectivity, or solubility issues to be investigated, as outlined in the discussion.

## 4. Discussion

From more extensive in vitro profiling of hit compounds from the MMV Pathogen Box [16] with activity against *T. cruzi* and/or *T.b. brucei* parasites, we have identified selective, fast-acting, efficacious and cidal compounds with activity against both parasite species. These compounds possess promising in vitro profiles warranting target identification and/or progression as hit molecules (Table 5). This is the first study that has classified the compounds from the Pathogen Box with respect to speed of action against these kinetoplastids, in addition to their static/cidal action and TcCYP51 activity against *T. cruzi*. Compounds with activity against TcCYP51 are generally deprioritised [25,27], as TcCYP51 inhibitors have failed to clear chronic Chagas disease infection in clinical trials [29]. Compounds need to be effective against small numbers of persistent parasites in the chronic stage of Chagas disease; thus, we also investigated the ability of compounds to have an irreversible effect against *T. cruzi* using a compound wash off assay, utilising 48 h incubation of *T. cruzi* in the presence compounds, followed by compound wash off, then 72 h incubation. Whilst this is an initial indication of an irreversible action on *T. cruzi* parasites, comparative studies need to be undertaken with a longer incubation time, in concert with in vivo studies. It is important to develop longer-term assays, where consideration is given to the addition of fresh host cells [49], that divide slowly to prevent overgrowth [26], and to incorporate monitoring of host cell health. Prior to time-kill assays, the cytotoxicity of the Pathogen Box compounds was assessed. We previously had determined the selectivity of the Pathogen Box compounds for *T. cruzi* and *T.b. brucei* in relation to HEK293 [16]. To facilitate a more accurate determination of the selectivity of the compounds, we sourced solid samples, enabling an increase in the top concentration tested from 5 µM to 20 µM. Compounds exhibiting a SI > 10 were prioritised for further investigation, as recommended by the Drugs for Neglected Diseases initiative (DND*i*) [25]. We have also identified slow-acting compounds against *T. cruzi* with activity against both parasites that are recommended for further investigation in the drug discovery pipeline.

Following in vitro profiling of the Pathogen Box collection, we identified 9 pan-active, selective, and fast-acting compounds (Table 5). These compounds did not inhibit TcCYP51, and 8 of these compounds showed no shift in the IC_50_ value against *T. cruzi* following compound wash off (<2-fold difference), thus suggesting no outgrowth following removal of compound pressure and a further incubation for 3 days. MMV652003 showed a 3.4-fold increase in the IC_50_ value following wash off, which may suggest outgrowth at some concentrations; however, this did not occur at the E_max_, which remained constant. This compound is a boron containing molecule of the benzoxaborole (oxaborole) chemical class [50]; however, it has been identified with in vitro and in vivo activity against both *T. cruzi* and *T.b. brucei*, and thus is not a new class to pursue. Through a prior medicinal chemistry campaign, MMV652003 (AN3520) led to the development of SCYX-6759 (AN4169) [44], with curative activity in a stage 2 mouse model of HAT (utilising *T.b. brucei*) and is now an advanced lead for Chagas disease [45]. Subsequently, SCYX-7158 (AN5568) was developed with improved pharmacokinetic properties against HAT spp, which is currently in ongoing clinical trials as an oral treatment of HAT caused by *T.b. gambiense* [42] and has shown 100% cure of mice infected with *T. cruzi* when treated for 40 days [4]. It has recently been shown, using live-cell imaging, that SCYX-6759 demonstrated a 20-h lag phase of inhibition following exposure of *T. cruzi* infected H9C2 rat heart cells to 16 µM of compound, suggesting a slow MOA [46]. However, the study did not state the IC_50_ value, nor the EC_100_. It may be that the rate of inhibition is relative to concentration, as has been previously suggested for this compound [32]. For *T. cruzi* therefore, longer term wash off assays would be warranted for this class of compound. Whilst the target of benzoxaboroles is known to be mRNA processing endonuclease, CPSF3 in *T.b. brucei* [47] the target of oxaboroles remains unknown for *T. cruzi*. Thus, it would be beneficial to identify the MOA/ target, to facilitate further optimisation [32].

The 2-aryl oxazole class of compounds (MMV688797, MMV688958 and MMV688795) from the Pathogen Box collection were fast-acting against *T.b. brucei* and *T. cruzi*, with cidal activity demonstrated against *T. cruzi*. We have previously identified the activity of 2-aryl oxazole (2-phenylthiazole) chemical class against *T.b. brucei* in 2012 [24]. Follow up chemical modifications resulted in new, more active compounds, including MMV688958, of which a related compound was fast-acting against *T. cruzi* parasites [48]. A related compound to MMV688958 has also demonstrated activity in a mouse model of *T. cruzi* infection, following pre-treatment of mice with the nonselective CYP-inhibitor, 1-aminobenzotriazole, potentially mediated by the thiazole phenyl substituent resulting from rapid metabolism (compound 64a [48]). Pathogen Box associated data indicated that the aryl oxazoles were active against *T.b. rhodesiense* [10], and we have identified that these three compounds were also active against *T.b. gambiense*. Improving the metabolic stability of aryl oxazoles is ongoing [48]. The dual inhibitory nature of this chemical class against kinetoplastids further supports these efforts. Additionally, we have shown that MMV688958 does not inhibit TcCYP51, an important attribute for ongoing development in the drug discovery pipeline.

MMV688796 is 2,4-substituted furan, belonging to the kinetoplastid compound set in the Pathogen Box collection, previously demonstrated to be active against *T.b. rhodesiense* [51]. We have demonstrated that MMV688796 also has activity against *T.b. gambiense* and is a promising, fast-acting hit compound against both parasites. The in vitro DMPK data provided with the MMV Pathogen Box collection shows MMV688796 has favourable Log D, suggesting good lipophilicity, in addition to a favourable LogP (PubChem, https://pubchem.ncbi.nlm.nih.gov/, accessed on 27 May 2022) with a moderate half-life of 3.9 h, when dosed orally in mice [51]. Therefore, MMV688796 was recommended for progression to in vivo mouse models of infection for *T.b. rhodesiense*, *T.b. gambiense* and *T. cruzi*.

MMV688550 is an imidazo [1,2] purine which originated from the Pathogen Box kinetoplastid set, with activity against *T.b. rhodesiense* [51] and now with confirmed activity against both *T.b. rhodesiense* and *T.b. gambiense* in the current study. MMV688550 did not display activity against the mammalian cells at the concentrations and incubation times tested, nor against TcCYP51, with a favourable half-life and in vitro DMPK [51], and thus warrants in vivo investigation.

MMV689028 is a fast-acting benzyl piperazine, which showed no activity against HEK293 and HEPG2, nor exhibited TcCYP51 activity. Both MMV689028 and the related compound, MMV689029, induced CYP1A2 activity, resulting in either high clearance or low half lives in vivo, provided in the Pathogen Box in vitro DMPK data [10]. The in vitro profile of MMV689028, with an improved half-life in comparison to MMV689029 (6.8 and 2.4 h, respectively), should be investigated for in vivo activity against *T. cruzi* and *T.b. rhodesiense*/*T.b. gambiense*, potentially with the use of a CYP inhibitor as a proof of concept, as this enzyme contributes to elimination of xenobiotics, which may influence drug clearance [52].

All fast-acting compounds that were identified originated from the Pathogen Box kinetoplastid set, whilst MMV687248 was from the tuberculosis set. This compound showed moderate selectivity to *T. cruzi* in relation to HEPG2 cells (SI = 13). Thus, selectivity would need to be improved, by sourcing and/or synthesising analogues [10]. The remainder of the tuberculosis set of compounds were either cytotoxic or did not confirm activity for one or more of the parasites, as per example, MMV687273 (SQ-109), currently in clinical trials for the treatment of tuberculosis, has previously reported activity against *T.b. brucei* with an IC_50_ value of 5 µM [53]. This compound had poor activity against *T.b. brucei* in our original testing and did not reconfirm activity against *T. cruzi*, as we originally identified [16]. Following 48 h an IC_50_ value of 13 µM was demonstrated against *T. cruzi* in our studies, with 16 µM, following 72 h incubation (results not shown). Previous studies have suggested that this was a lead compound against *T. cruzi* [54], with an IC_50_ value of 0.5–1 µM, following 96 h incubation. The incubation time in the studies would need to be extended to 96 h and beyond to confirm this; however, it may be that the image-based format used for our analysis is more sensitive and thus identifies small numbers of parasites indicative of infected cells, in comparison to Giemsa staining [54]. This compound was therefore of limited interest for both parasites.

We identified whether some of the slower-acting compounds against *T. cruzi*, with efficacy against both parasites had potential with respect to further drug discovery efforts. The majority of these have lack of selectivity to one mammalian cell line, and therefore SAR is recommended to identify more selective molecules. Unlike many TcCYP51 inhibitors which are typically slow-acting against *T. cruzi* [23], MMV028694, MMV688372, MMV688283, MMV687776 and MMV688467 had high efficacy (>95%) against *T. cruzi* following 48 h incubation. The phenotypic profile suggests these compounds may not be TcCYP51 inhibitors [23], which was confirmed by testing against recombinant TcCYP51. MMV028694, a 2,4-disubstituted pyrimidine showed moderate and slow-acting activity against *T. cruzi* following 48 h incubation (9.4 µM) and 72 h incubation (16 µM) with poor selectivity toward *T. cruzi* in relation to HEK293 cells (SI = 1); however, it displayed selectivity to 3T3 fibroblasts (SI > 7.8). This compound showed moderate selectivity to *T.b. brucei* (SI = 12), as well as activity against both infective forms of *T.b. brucei*. However, medicinal chemistry efforts to improve the moderate activity and selectivity observed are required before further progression of this compound would be considered. MMV688467 was fast-acting against *T.b. brucei* but slow-acting against *T. cruzi*. This compound is a butyl sulphanilamide showed activity against HEPG2 cells, resulting in a SI to *T. cruzi* of 10 and for *T.b. brucei*, 74. This compound could be pursued for *T.b. brucei* but would need to be tested against the infective forms of the parasite. More selective analogues would be required to be considered for further studies against *T. cruzi.*

MMV688372 is a substituted 2-phenylimidazopyridine which is slow-acting against *T. cruzi*, with a 13-fold difference in the IC_50_ value from 24 to 72 h incubation. Whilst an IC_50_ value could be estimated following 24 h incubation, this was sub- efficacious (75% E_max_ against *T. cruzi*). This compound shows a lower SI against *T. cruzi* in relation to HEK293 and HEPG2 cells, with an SI of 14 and 129 for *T. cruzi* and *T.b. brucei*, respectively, in relation to HEK293 cells, and an SI of 28 for *T. cruzi* and 257 for *T.b. brucei* in relation to HEPG2 cells. MMV688372 has recently been shown to have activity against the protozoan parasite that causes East Coast fever in cattle, *Theileria parva* [55], with a low µM IC_50_ value (5.3 µM) against bovine peripheral blood mononuclear cells (PBMCs), following 24 h incubation [55]. MMV688372 also has associated CYP (1A2; 2C9 and 2D6) activity; thus, a CYP inhibitor may be required for in in vivo studies, to provide proof of concept. However, improving the current moderate selectivity against *T. cruzi* would need to be explored with generation of analogues. MMV688283 is a 4-amino quinoline, with moderate activity against both parasites, slow-acting activity against *T. cruzi* and limited selectivity. Synthesis of analogues is also required to ascertain structure activity relationships to improve moderate activity before proceeding further. The benzoxaborole MMV687776 was slow-acting against *T. cruzi*, with moderate activity displayed against *T.b. brucei* following 72 h incubation (4.3 µM) and low selectivity to *T.b. brucei* and *T. cruzi* to HEK293 (SI = 3.0, 8.1, respectively). Therefore, this was a different activity profile to the fast-acting benzoxaborole (MMV652003) identified. However, it had good selectivity for *T.b. brucei* in relation to HEPG2 (SI > 18), and *T. cruzi* in relation to 3T3 cells (SI > 46). Further profiling with other mammalian cells would be warranted and potentially SAR to improve potential cytotoxicity. However, benzoxaborole compounds are oxaborole pharmacophores [56], and oxaboroles are well-developed leads against both *T.b. brucei* and *T. cruzi* [50]. As this class is not novel, and the target is known for *T.b. brucei*, it would be beneficial to identify the biological target of this compound class in *T. cruzi*, rather than development of this compound for *T.b. brucei*. A recent study showed that carbonic anhydrases from *T. cruzi* and *L. donovani chagasi* are inhibited by benzoxaboroles. However, whilst there was low µM activity against *L. donovani*, there was less activity (13–88 µM) shown against the *T. cruzi* carbonic anhydrase [57]; thus, further work is needed to confirm the target. The activity of this and other slow-acting compounds would also need to be tested against the human infective forms of HAT.

The pyrazolopyridine MMV099637 with an IC_50_ value of 2.7 µM against *T. cruzi* (2.1 µM against *T.b. brucei*) was not previously active against *T. cruzi* at a final assay concentration of 20 µM [16]. Stability of some compounds in DMSO, due to various factors such as storage and freeze-thaw cycles [58], may have contributed to variation in activity. MMV099637 was slow-acting with low efficacy and whilst low efficacy is commonly associated with TcCYP51 activity [23], this was not the case with this compound. MMV099637 may present with this profile due to solubility issues noted during preparation of DMSO stock solutions (maximum 14 mM due to solubility issues in 100% DMSO); thus, in vitro solubility analysis, such as a nephelometry assay, to confirm if solubility is an issue, should be undertaken [59]. MMV688776 is a pyrazoloquinazoline with moderate activity against *T. cruzi* (3.1 µM) and sub-µM activity (0.79 µM) against *T.b. brucei*, which also was a sub-efficacious compound against *T. cruzi*. This compound did not inhibit TcCYP51 and may present with this profile due to solubility issues noted during preparation of DMSO stock solutions (maximum 14 mM due to solubility issues in 100% DMSO). Whilst pyrazoloquinazolines do not have reported activity against trypanosomes, compounds from this class have demonstrated inhibition of human topoisomerase I [60]. Topoisomerase has been identified as a target in trypanosomes and has recently been implicated in response to replication stress in *T. cruzi* [61].

Whilst we have identified compounds that are selective and pan-active against *T. cruzi* and *T.b. brucei* parasites, we also characterised these compounds based on their speed of action. While fast-acting compounds may be beneficial down the pipeline, this is not necessarily a prerequisite when considering the progression of compounds. However, a fast-acting and long-lasting compound is recommended by DND*i* as optimal for Chagas disease, as it is questionable whether a slow-acting compound is desirable due to the clinical failure of slow-acting azoles [25]. Thus, fast-acting compounds are often prioritised in drug discovery campaigns [46]. Since the fast- and slow-activity compounds identified herein showed activity against *T. cruzi* and *T.b. brucei*, target-based studies could be undertaken with *T.b. brucei* parasites, as these do not require a host cell and thus are cultured axenically. Ultimately, thorough in vitro evaluation of these compounds provided a valuable resource for researchers and importantly, provided compounds for further pre-clinical and target-based studies to elucidate potentially new modes of action in both *T. cruzi* and *T.b. brucei*, thereby supporting the open-source initiative for the Pathogen Box collection.

## Figures and Tables

**Figure 1 microorganisms-10-01287-f001:**
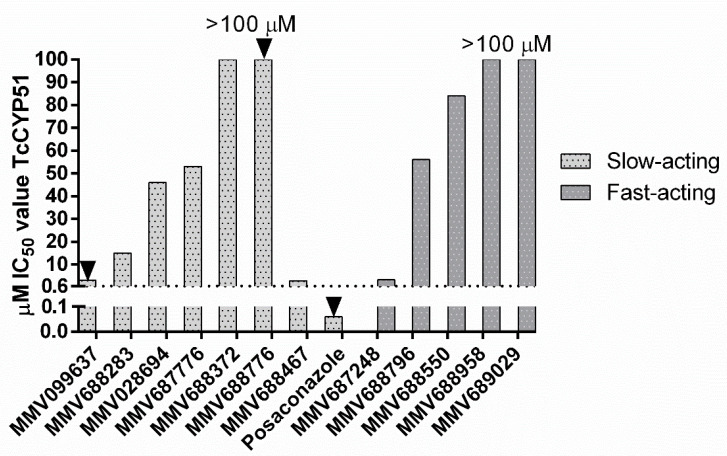
IC_50_ values of slow and fast-acting compounds active against TcCYP51. Where IC_50_ values reach 100 µM, the IC_50_ value is >100 µM. Arrows denote compounds with a sub-efficacious E_max_ following 48 h incubation. MMV099637 = 70% E_max_; MMV688776 = 88%, Posaconazole = 73% E_max_. Dashed line = 10× the IC_50_ value of posaconazole as a cut off for CYP51 inhibition.

**Table 1 microorganisms-10-01287-t001:** Activity of compounds with activity against both *T.b. brucei* and *T. cruzi* parasites.

Compound	IC_50_ (*T.b. brucei*) (E_max_ %)	Chemical Class	IC_50_ (*T. cruzi*) (E_max_ %)	IC_50_ HEK ^2^ (HEP) ^1^	SI (*T.b. brucei*) HEK (HEP)	SI (*T. cruzi*) 3T3 ^3^	SI (*T. cruzi*) HEK (HEP)
MMV 688372	0.014 ± 0.0024 (99)	substituted 2-phenylimidazopyridine	0.13 ± 0.0097 (99)	1.8 ± 0.0502 (3.6 ± 0.54)	129 (257)	>561	14 (28)
MMV 688550	0.025 ± 0.00083 (99)	imidazo [1,2] purine	0.53 ± 0.110 (100)	NA (NA)	>3160 (>3160)	>138	>149 (>149)
MMV 652003	0.092 ± 0.0027 (100)	benzoxaborole	0.15 ± 0.004 (100)	6.8 ± 0.204 (50%)	74 (>862)	>487	45 (>548)
MMV 688797	0.097 ± 0.0021 (99)	2-aryl oxazole	0.51 ± 0.11 (100)	NA (NA)	>814 (>814)	>143	>154 (>154)
MMV 688795	0.13 ± 0.052 (99)	2-aryl oxazole	0.58 ± 0.0503 (100)	NA (NA)	>608 (>608)	>126	>136 (>136)
MMV 688796	0.12 ± 0.023 (99)	2,4-substituted furan	1.2 ± 0.4 (100)	NA (NA)	>658 (>658)	>61	>66 (>66)
MMV 689028	0.12 ± 0.030 (99)	benzyl piperazine	0.70 ± 0.13 (100)	NA (NA)	>658 (>658)	>104	>113 (>113)
MMV 688958	0.14 ± 0.028 (99)	2-aryl oxazole	0.98 ± 0.11 (100)	NA (NA)	>564 (>564)	>75	>81 (>81)
MMV 688776	0.79 ± 0.050 (99)	pyrazoloquinazoline	3.1 ± 0.73 (87) ^5^	NA (NA)	>100 (>100)	>24	>25 (>25)
MMV 688467	0.31 ± 0.0069 (99)	butyl sulfanilamide	2.3 ± 0.46 (98)	102% (23 ± 3.8)	>254 (74)	>32	>34 (10)
MMV 689029	0.42 ± 0.053 (99)	benzyl piperazine	1.2 ± 0.067 (100)	NA (NA)	>188 (>188)	>61	>66 (>66)
MMV 687248	0.51 ± 0.045 (99)	3,5-disubstituted pyridine	1.5 ± 0.13 (99)	NA (19.6 ± 0.083)	>155 (38)	>49	53 (13)
MMV 099637	2.13 ± 0.064 (99)	pyrazolopyridine	2.7 ± 0.41 (70) ^5^	NA (NA)	>37 (>37)	>27	>29 (>29)
MMV 028694	0.83 ± 0.057 (100)	2,4-disubstituted pyrimidine	9.4 ± 1.8 (98)	9.7 ± 0.050 (NT)	12	>7.8	1.0 (NT)
MMV 687776	4.3 ± 1.3 (99)	benzoxaborole	1.6 ± 0.38 (100)	13 ± 1.6 (89%)	3.0 (>18)	>46	8.1 (>50)
MMV 688283	7.6 ± 0.27 (99)	4-amino quinoline	6.6 ± 1.2 (100)	100% (109%)	>10 (>10)	>11	>12 (>12)
MMV 006901	4.9 ± 0.0085 (99)	2,4-aminoquinoline	1.2 ± 0.11 (100)	18 ± 0.81 (9.8 ± 1.5)	3.7 (1.8)	>60 ^4^	15 (7.1)
MMV 22029	1.5 ± 0.059 (99)	biaryl sulfonamide	8.2 ± 0.69 (100)	8.1 ± 0.054 (NT)	5.6	1.2	1.0 (NT)
MMV 688514	1.50 ± 0.027 (99)	ND	4.9 ± 0.16 (100)	9.1 ± 0.46 (NT)	6.0	2.7	1.9 (NT)
MMV 22478	1.5 ± 0.18 (98)	pyrazolo[1,5-a]pyrimidine	5.9 ± 1.3 (100)	12 ± 0.39 (NT)	8.0	1.04	2.03 (NT)
MMV 21013	2.2 ± 0.088 (100)	2-pyridyl-4-aminopyrimidine	1.8 ± 0.49 (100)	4.0 ± 0.22 (106%)	1.8 (>36)	>41 ^4^	2.2 (NT)
MMV 687706	2.2 ± 0.0028 (99)	1-[3-(4-phenoxyphenyl)-1H-pyrazol-5-yl]piperazine	11 ± 0.205 (99)	16.2 ± 5.2 (NT)	7.4	>6.6	1.5 (NT)
Pentamidine	0.0031 ± 0.0001 (100)	aromatic diamidine	NT	NA at 0.67 µM (NT)	>216	NT	NT
Diminazene aceturate	0.037 ± 0.00041 (100)	phenylhydrazine	NT	NA at 38 µM (NT)	>1027	NT	NT
Amphotericin B	1.5 ± 0.088 (100)	polyene macrolide	NT	NA at 4.0 µM (NT)	>2.6	NT	NT
Posaconazole	NT	triazole	0.0055 ± 0.0000041 (77) ^5^	NT (NA at 1 µM)	NT	>333	NT
Benznidazole	NT	nitroheterocyclic	3.4 ± 0.71 (100)	NT (NA at 127 µM)	NT	>37	NT
Nifurtimox	NT	nitroheterocyclic	0.85 ± 0.048 (100)	NT (74% at 127 µM)	NT	>149	NT
Puromycin	0.063 ± 0.00041 (100)	peptidyl nucleoside	5.6 ± 0.011 (100)	0.73 ± 0.27 (0.25 ± 0.015)	12 (4.0)	1.2	0.13 (0.044)

^1^ HEP = HEPG2 (human hepatoma nontumorigenic cells with epithelial-like morphology). ^2^ HEK = HEK293 (human embryonic kidney cells). ^3^ 3T3 = mouse embryonic fibroblast cells. ^4^ Activity against 3T3 cells replicate 1 = 1.2 µM, replicate 2 did not reach plateau of activity. ^5^ Sub-efficacious against *T. cruzi*. NT = Not tested. NA = Not active. IC_50_ value cannot be determined. Where activity is >50% at the highest dose, the % activity is shown. Highest dose for HEPG2 and HEK293 cells = 79 µM, 3T3 = 73 µM.

**Table 2 microorganisms-10-01287-t002:** IC_50_ values and maximum activity (E_max_) of compounds with activity against *T. cruzi* and *T.b. brucei* following 24–72 h exposure; in addition to compound wash off in the *T. cruzi* assay.

Compound	IC_50_ 24 h *T. cruzi* (E_max_)	IC_50_ 72 h *T. cruzi* (E_max_)	IC_50_ Wash Off (E_max,_ Fold Change ^1^)	IC_50_ 24 h *T.b. brucei* (E_max_)	IC_50_ 48 h *T.b. brucei* (E_max_)	Set ^2^	Speed of Action ^3^
MMV 688550	0.90 ± 0.17 (80)	0.44 ± 0.089 (100)	0.31 ± 0.067 (100; 0.58)	0.022 ± 0.014 (100)	0.021 ± 0.00053 (99)	Kinetoplastid	FA Cidal
MMV 652003	1.28 ± 0.56 (93)	0.28 ± 0.18 (100)	0.51 ± 0.080 (99; 3.4)	0.45 ± 0.067 (100)	0.11 ± 0.012 (100)	Kinetoplastid	FA Static^4^
MMV 688797	0.89 ± 0.135 (86)	0.43 ± 0.042 (99)	0.38 ± 0.056 (99; 1.3)	0.15 ± 0.029 (99)	0.10 ± 0.0062 (99)	Kinetoplastid	FA Cidal
MMV 688795	1.3 ± 0.12 (89)	0.62 ± 0.05 (99)	0.38 ± 0.056 (99; 0.74)	0.29 ± 0.0047 (99)	0.106 ± 0.020 (99)	Kinetoplastid	FA Cidal
MMV 688796	2.4 ± 1.3 (91)	1.02 ± 0.057 (100)	1.1 ± 0.48 (98; 1.1)	0.24 ± 0.053 (82)	0.097 ± 0.0048 (100)	Kinetoplastid	FA Cidal
MMV 689028	1.1 ± 0.43 (91)	0.52 ± 0.13 (100)	0.39 ± 0.020 (99; 0.86)	0.72 ± 0.38 (78)	0.11 ± 0.0032 (99)	Kinetoplastid	FA Cidal
MMV 688958	1.74 ± 0.18 (94)	0.601 ± 0.093 (100)	0.601 ± 0.093 (100; 0.73)	0.201 ± 0.028 (78)	0.14 ± 0.017 (99)	Kinetoplastid	FA Cidal
MMV 689029	3.7 ± 0.26 (90)	1.3 ± 0.28 (99)	0.88 ± 0.056 (98; 0.73)	0.98 ± 0.34 (89)	0.40 ± 0.023 (100)	Kinetoplastid	FA Cidal
MMV 687248	4.1 ± 1.3 (100)	3.5 ± 0.16 (100)	2.0 ± 0.039 (100; 1.3)	0.93 ± 0.047 (87)	0.2 ± 0.032 (99)	Tuberculosis	FA Cidal
MMV 688372	1.6 ± 0.064 (75)	0.12 ± 0.052 (100)	0.13 ± 0.0029 (100; 0.86)	0.034 ± 0.0020 (83)	0.013 ± 0.00032 (100)	Kinetoplastid	SA (*TC*) Cidal
MMV 688467	4.4 ± 0.73 (75)	2.5 ± 1.4 (99)	2.10 ± 0.44 (100; 0.91)	1.2 ± 0.35 (88)	0.46 ± 0.018 (100)	Kinetoplastid	SA (*TC*) Cidal
MMV 028694	58% at 73 µM	16 ± 1.70 (100)	12 ± 1.8 (98; 1.2)	0.70 ± 0.19 (87)	0.71 ± 0.13 (100)	Malaria	SA (*TC*) Cidal
MMV 687776	94% at 73.3 µM	1.3 ± 0.30 (99)	1.6 ± 0.064 (100; 1.0)	5.35 ± 0.103 (89)	4.04 ± 1.78 (99)	Lymphatic filiariasis	SA (*TC*) Cidal
MMV 688283	58% at 73 µM	7.9 ± 0.51 (96)	7.6 ± 1.5 (100; 1.2)	7.8 ± 0.32 (94)	6.2 ± 0.45 (96)	Kinetoplastid	SA (*TC*) Cidal
MMV 688776	82% at 49 73 µM	2.43 ± 0.023 (820)	3.5 ± 0.101 (87; 1.1)	2.3 ± 0.17 (60)	0.86 ± 0.14 (99)	Kinetoplastid	SA (*TC, TBB*) Not efficacious (*TC*) ^4^
MMV 099637	NA at 73 µM	2.41 ± 0.070 (65)	3.3 ± 0.045 (92; 1.2)	2.6 ± 0.11 (50)	2.2 ± 0.13 (99)	Kinetoplastid	SA (*TC*, *TBB*) Not efficacious (*TC*) ^4^
Pentamidine	NT	NT	NT	0.012 ± 0.0021 (99)	0.0002 ± 0.0000021 (100)	-	FA Cidal
Diminazene aceturate	NT	NT	NT	0.21 ± 0.058 (99)	0.070 ± 0.027 (100)	-	FA Cidal
Amphotericin B	NT	NT	NT	0.88 ± 0.163 (100)	0.26 ± 0.0016 (100)	-	FA Cidal
Posaconazole	NA at 1 µM	0.0035 ± 0.000065 (97) ^4^	0.0018 ± 0.000078 (100; 0.36)	NT	NT	-	SA Not efficacious (*TC*) ^4^
Benznidazole	4.7 ± 1.4 (100)	5.6 ± 0.0021 (100)	1.2 ± 0.11 (100; 0.29)	NT	NT	-	FA Cidal
Nifurtimox	1.03 ± 0.076 (100)	0.85 ± 0.088 (100)	0.35 ± 0.014 (100; 0.60)	NT	NT	-	FA Cidal
Puromycin	3.5 ± 0.44 (98101)	2.55 ± 0.47 (100)	2.8 ± 0.13 (100; 0.50)	0.096 ± 0.013 (100)	0.097 ± 0.00042 (100)	-	FA Cidal

^1^ Fold change is the difference between the IC_50_ value from 48 h incubation in the *T. cruzi* assay to the IC_50_ value following wash off and incubation for a further 72 h. A >2-fold difference in the IC_50_ value in comparison to 48 h incubation following removal of compound, defined as static activity. ^2^ Set = classification of compounds provided in the Pathogen Box. ^3^ Speed of action. Slow-acting (SA) = IC_50_ value unable to be determined following 24 h incubation (no plateau of activity, or plateau of activity is <80% or IC_50_ value < 10 µM), or >10-fold difference in the IC_50_ value from 24 to 72 h. For 72 h incubation against *T.b. brucei* see Table 1. Fast-acting (FA) = IC_50_ value can be determined following 24 h incubation, <10-fold difference in the IC_50_ value from 24 to 72 h. MMV689028 and MMV688958 considered FA with an EC_100_ of 78% against *T.b. brucei* following 24 h. ^4^ Not efficacious *T. cruzi* = E_max_ < 90% following 48 h incubation, see Table 1. TBB = *T.b. brucei*, TC = *T. cruzi*.

**Table 3 microorganisms-10-01287-t003:** Activity against the human infective forms of HAT, *T.b. gambiense* and *T.b. rhodesiense* of fast-acting MMV Pathogen Box compounds active against *T.b. brucei* and *T. cruzi*.

Compound	Chemical Structure	Chemical Class	IC_50_ *T. bR1* ^1^	IC_50_ *T. Bg* ^2^ STIB930	IC_50_ *T.b. brucei* ^3^
MMV 688550	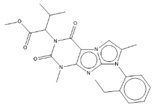	imidazo [1,2] purine	0.080 ± 0.0014	<0.001	0.025 ± 0.00083
MMV 652003 ^1^	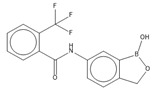	benzoxaborole	NT	NT	0.092 ± 0.0027
MMV 688797 ^2^	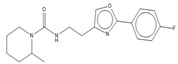	2-aryl oxazole	NT	NT	0.097 ± 0.0021
MMV 688795	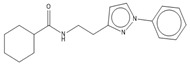	2-aryl oxazole	0.079 ± 0.00070	0.63 ± 0.0014	0.13 ± 0.052
MMV 688958	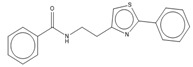	2-aryl oxazole	0.067 ± 0.0028	0.65 ± 0.050	0.14 ± 0.028
MMV 688796	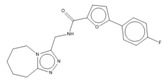	2,4-substituted furan	0.046 ± 0.031	0.16 ± 0.011	0.12 ± 0.023
MMV 689028	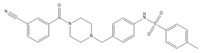	benzyl piperazine	0.28 ± 0.011	0.048 ± 0.0011	0.12 ± 0.030
MMV 689029	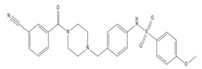	benzyl piperazine	0.63 ± 0.11	0.40 ± 0.12	0.42 ± 0.053
MMV 687248	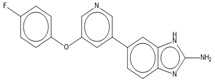	3,5-disubstituted pyridine	1.2 ± 0.54	0.24 ± 0.022	0.51 ± 0.039
Melarsoprol	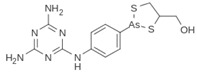	substituted aniline	0.015 ± 0.0040	0.011 ± 0.017	0.0031 ± 0.00019
Pentamidine	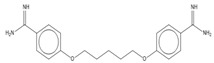	aromatic diamidine	NT	0.0030 ± 0.0020	NT

^1^ MMV652003 not tested as this class of compound (benzoxaborole) has already been identified with activity against *T. brucei* spp. [42]. ^2^ MMV688797 not tested as two analogues from the 2-aryl oxazole class were included. ^3^ IC_50_ values against *T.b. brucei* are derived from Table 1. TbR= *Trypanosoma brucei rhodesiense.* TbG = *Trypanosoma brucei gambiense.* DMPK = Drug metabolism and pharmacokinetics. PK= pharmacokinetics. SAR = Structure-activity relationship. NT = Not tested.

**Table 4 microorganisms-10-01287-t004:** Physicochemical properties of fast-acting Pathogen Box compounds with activity against *T.b. brucei* and *T. cruzi*.

Compound	Mwt	LogP ^1^	Pathogen Box Set	Commercially Available
MMV 688550	437.49	4.6	Kinetoplastid	No
MMV 652003	321.06	NC	Kinetoplastid	No
MMV 688797	331.38	3	Kinetoplastid	Yes
MMV 688795	297.39	3.6	Kinetoplastid	No
MMV 688796	354.38	2.5	Kinetoplastid	Yes
MMV 689028	474.57	3.3	Kinetoplastid	No
MMV 688958	308.40	3.9	Kinetoplastid	Yes
MMV 689029	449.62	2.9	Kinetoplastid	No
MMV 687248	320.32	3.3	Tuberculosis	No
Melarsoprol	398.341	1.3	Not in set (control)	Yes
Pentamidine	340.42	2.3	Not in set (control)	Yes

NC = no computed physicochemical properties listed on PubChem. ^1^ XLogP for MMV compounds sourced from PubChem. LogP for control drugs sourced from DrugBank.

**Table 5 microorganisms-10-01287-t005:** Prioritised compounds from the Pathogen Box with activity against both *T. cruzi* and *T. brucei* recommended for progression.

Compound/s	Chemical Class	Speed of Action	Potential Liability	Recommendation	References	New Class
MMV652003	benzoxaborole	Fast	Static action ^1^	Identify target/s (*T. cruzi*)	[4,42,44,45,46,47]	No
MMV688797 MMV688958 MMV688795	2-aryl oxazole	Fast	Metabolic Stability ^2^	Identify target/s Improve PK ^2^	[24,48]	No
MMV688796	2,4-substituted furan	Fast	-	In vivo studies	-	Yes
MMV688550	imidazo [1,2] purine	Fast	DMPK ^3^	In vivo studies ^4^ Improve PK ^3^	-	Yes
MMV689028 MMV689029	benzyl piperazine	Fast	DMPK ^3^	In vivo studies ^4^ Improve PK ^3^	-	Yes
MMV687248	3,5-disubstituted pyridine	Fast	Selectivity *T. cruzi* ^1^	Improve moderate selectivity (SAR)	-	Yes
MMV688372	substituted 2-phenylimidazopyridine	Slow	Selectivity *T. cruzi* ^1^	Improve moderate selectivity (SAR)	-	Yes

^1^ Potential liabilities identified from in vitro profiling. ^2^ Potential liabilities (of compound class) identified from the literature. ^3^ Potential liabilities identified from DMPK data provided with Pathogen Box [10]. ^4^ In vivo studies may require a CYP1A2 or CYP2C9 inhibitor. DMPK = drug metabolism and pharmacokinetics. PK = pharmacokinetics. SAR = structure-activity relationship. References provided if the compound class has been characterised with in vitro/in vivo activity against both parasites.

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
