# Peer review of "Temporal and Wash-Out Studies Identify Medicines for Malaria Venture Pathogen Box Compounds with Fast-Acting Activity against Both Trypanosoma cruzi and Trypanosoma brucei"

_microorganisms, 2022, doi:10.3390/microorganisms10071287_

Round 1

Reviewer 1 Report

The manuscript is well written and highlights the limitations and potential of the study. I recommend publication in this journal.

Author Response

The authors thank the reviewer for their comments.

Reviewer 2 Report

Manuscript entitled ¨Temporal and wash-out studies identify Medicines for Malaria Venture Pathogen Box compounds with fast-acting activity against both Trypanosoma cruzi and Trypanosoma brucei¨ by Sykes and co-workers . reports molecules with pharmacological potential against T. cruzi and T. brucei. Considering the urgent need for the identification of molecules with trypanocidal potential, the work is highly relevant.  

Before the article is published, it is necessary that the authors take into account the following specific considerations:

Authors must review all scientific nomenclature (Example: ¨shistosoma¨). (line 82) Please write all scientific names in italics

How was the recombinant CYP51 enzyme obtained? Was recombinant expression performed? What was the procedure to determine the inhibitory properties of the molecules? Please provide more details of the methodology of those experiments.

Please reduce the amount of methodological details in this section. Write these details in the methodology section. Here, specify only the results obtained (Line 488)

I did not find the ¨table 6¨ (Line 538)

Is it possible to propose any mechanism of inhibition of the CYP51 enzyme?

Please check the reference list (format and scientific names)

Author Response

The authors sincerely thank the reviewer for their comments and questions.

Response: On line 85, Shistosoma is in italics, however the genus name Babesia was not. This has been changed (line 85). Other genus/ species names in the document are italicised.

How was the recombinant CYP51 enzyme obtained? Was recombinant expression performed? What was the procedure to determine the inhibitory properties of the molecules? Please provide more details of the methodology of those experiments.

Please reduce the amount of methodological details in this section. Write these details in the methodology section. Here, specify only the results obtained (Line 488)

Response: Some of the information from the results section has been incorporated  into the methodology section describing compound testing against TcCYP51. Other parts have now been incorporated into the discussion. Have added reference to this section which describes development of the TcCYP51 assay this includes how was the recombinant CYP51 enzyme was obtained, recombinant expression and the assay procedure.

I did not find the ¨table 6¨ (Line 538)

Response: This was meant to be Table 5. This has been modified. This is now line 534.

Is it possible to propose any mechanism of inhibition of the CYP51 enzyme?

Response: In terms of the TcCYP51 assay, this identifies direct binding to TcCYP51. This mechanism is where compounds directly bind to the active site, thus competitively bind to the haem within CYP51, preventing the substrate from binding. Commonly, a basic atom from a heterocyclic ring interacts with the haem iron to occupy the active site. Ketoconazole, fluconazole, itraconazole, ravuconazole and posaconazole have been shown to have this MOA, in vitro.

Other CYP51 mechanisms include inhibition of the endogenous substrate lanosterol, however this has not been determined in the TcCYP51 assay, as this would require mass spectrometry and significantly limited throughput.

Please check the reference list (format and scientific names)

Response: Since Microorganisms allows “Your references may be in any style, provided that you use the consistent formatting throughout. It is essential to include author(s) name(s), journal or book title, article or chapter title (where required), year of publication, volume and issue (where appropriate) and pagination”. We have utilised the Endnote Style of PLOS Neg. Trop. Diseases. This does not italicise scientific names. Microorganisms does encourage having the DOI with references, therefore the DOI has been added to the citation style in Endnote.

Editorial correction. In the graph for TcCYP51 inhibition, there was a “0” missing in MMV082694. This has been changed in the graph. MMV688776 was missing an arrow to designate a sub-efficacious effect (shown in Table 2, footnote 4).

Reviewer 3 Report

After a detailed evaluation of the scientific findings, I think the manuscript presents important data about the potential activity on trypanosomes. I only think these analysis should be included together with the first description of trypanocidal activity or associated with the further experiments of the mechanism of action. The publication of these data separate of the next steps, looks like a data segmentation.

Author Response

Response: The authors thank the reviewer for their comments.

This article is the basis for further studies, however these have not yet been undertaken. It is important to deliver the in vitro prioritisation of these compounds to give other researchers already working on the Pathogen Box this information, to support open-source research and provide others the opportunity to work on these compounds with the knowledge provided. We have stated that there was a previous paper which was the basis for these studies. Please see line 89, reference 16.

Reviewer 4 Report

Abstract

line 24: time to kill: this a movie name. do you mean IC50?

Introduction:

Line 101: the speed of compound action.? What does this phrase mean?

Author Response

Response: We thank the reviewer for their comments.

Time to kill assays have been changed to time-kill to void any further confusion. Time-kill is widely used to describe the rate at which compounds inhibit target cells in vitro, and is often utilised for compound prioritisation. The criteria for activity at each time point is described in the methods and shown in Table 2.

Changes made to this in the abstract, lines 115, 288, 426.

Response: The speed of action is the rate of action of compounds against intracellular T. cruzi, determined from temporal (time-kill assays). The header for Table 2 has been modified to describe that the table shows the IC50 values and the maximum achievable activity of compounds, over time.

In the abstract, speed of action has been described “temporal and wash off assays were utilised to identify the speed of action and cidality of compounds”. The speed of action of compounds in in vitro assays is widely used to describe the time that it takes for compounds to be active against cells/ parasites/ bacteria. The criteria used to define an active compound, the speed of action and the cidality of compounds against T. cruzi are shown in Table 2.